# Depression, anxiety, and stress among frontline health workers during the second wave of COVID-19 in southern Vietnam: A cross-sectional survey

Anh Le Thi Ngoc[1], Chinh Dang Van[1], Phong Nguyen Thanh[2], Sonia Lewycka[3,4], Jennifer Ilo Van Nuil [4,5]*

1 Institute of Public Health, Ho Chi Minh City, Vietnam, 2 Hospital for Tropical Diseases, Ho Chi Minh City, Vietnam, 3 Oxford University Clinical Research Unit, Hanoi, Vietnam, 4 Nuffield Department of Medicine, Centre for Tropical Medicine and Global Health, University of Oxford, Oxford, United Kingdom, 5 Oxford University Clinical Research Unit, Ho Chi Minh City, Vietnam

* jvannuil@oucru.org

**Data Availability Statement:** Data are within the paper and its Supporting information files.

## Abstract

Health workers around the world have taken on massive frontline roles in the fight against COVID-19, often under intense pressure and in the face of uncertainty. In this study, we determined the rates of depression, anxiety, stress and related factors among health workers in COVID-19 designated hospitals in southern Vietnam during the second wave of COVID-19. From July-September 2020, we collected self-administered surveys from 499 health workers in 14 hospitals that were designated for the care and treatment of patients with COVID-19. The survey included sections on demographics, co-morbid health conditions, symptoms experienced during patient care, a depression, anxiety and stress assessment (DASS-21), and other related factors. We used logistic regression models to identify factors associated with depression, anxiety and stress, and adjusted for confounding factors. 18%, 11.5%, 7.7% of participants had symptoms of depression, anxiety, and stress, respectively with the majority at mild and moderate levels. The risk factors for increased mental health impact included long working hours, experiencing physical symptoms, fear of transmission to family, COVID-19 related stigma, and worry when watching media about COVID-19. Psychological counseling and training in infection prevention were protective factors that reduced the risk of mental health problems. Further exploration of the association between physical symptoms experienced by health workers and mental health may guide interventions to improve health outcomes. More routine COVID-19 testing among health workers could reduce anxieties about physical symptoms and alleviate the fear of transmitting COVID-19 to family and friends. Medical institutions need to ensure that health workers have access to basic trainings prior to initiation of work, and mental health support during the pandemic and into the future.

**Funding:** The study was funded by South Asian Field Epidemiology and Technology Network, (SAFETYNET), and the Field Epidemiology Training Program (FETP) in Vietnam (grant recipient: Ms Anh Le Thi Ngoc). The funders had no role in study design, data collection and analysis, decision to publish, or preparation of the manuscript.

**Competing interests:** None of the authors have competing interests to disclose.

## Introduction

Since the start of the COVID-19 pandemic in December 2019, health workers around the world have taken on massive frontline roles in the fight against COVID-19, often under intense pressure and in the face of uncertainty. We know from research on other viral epidemics (e.g. severe acute respiratory syndrome (SARS), Ebola) and more recent work in the context of COVID-19, that frontline workers experience an increased risk of both short and long-term psychological impacts, including depression, anxiety, stress, and Burnout Syndrome (BOS), among others [1, 2]. In an umbrella systematic review with data from previous epidemics, about one third of healthcare workers had BOS but data for COVID-19 were insufficient at that time [3]. However, a later review found high levels of anxiety, depression, BOS, and post-traumatic stress [2]. In early stages of the pandemic, for example, in China from January-February 2020, researchers found that about half (50.4%) of frontline health workers had symptoms of depression and 71.5% had symptoms of distress [4]. By mid-2020, the situation was largely unchanged. For example, in Malaysia, depression, anxiety, and stress scores remained high several weeks after lockdown was lifted, with fear of exposure to patients with COVID-19 increasing the likelihood of depression [5]. Longitudinal research in a hospital setting in Italy where there was constant influx of COVID-19 patients found an increase in mental health impacts over time from April 2020, December 2020, and May/June 2021, with depression rates rising from 51.1% to over 60% in December 2020, and up to 64% by May/June 2021 [6–8].

The risk factors associated with mental health outcomes are variable and context specific. Based on results from a rapid systematic review, in 17 or the 55 articles reviewed, fear was noted as a primary stressor: fear of infection, fear of transmitting to family and friends, fear of the unknown [1]. Linked to these fears, was also the impact of stigma on mental health outcomes, which was noted in SARS in 2003 [9] but also more recently during COVID-19 as a risk factor for experiencing depressive symptoms [10]. One study conducted in Italy during the peak of COVID-19 found that risk factors for stress included female gender, direct patient care, the need for social support, and avoidance strategies, while a positive attitude was protective [11]. In another study during the first wave in Italy, researchers found that while the prevalence of mental health issues was not larger than pre-pandemic prevalence in similar populations, those who had tested positive and those who had been exposed to COVID-19 were at increased risk of anxiety and depression [12]. Other systems-related factors that may influence the wellbeing of health workers included excessive working hours, lack of protective equipment and training on its use, and the importance of support systems (both personal and institutional) [1].

We designed a study to explore how COVID-19 impacts the wellbeing of healthcare professionals in Southern Vietnam to explore the factors that may lead to improved mental health outcomes. The protection of healthcare professionals is essential and understanding the factors that influence mental health can help us to provide immediate support, as well as better prepare for future pandemics.

## Methods

The objectives of the study were to determine the prevalence of depression, anxiety, and stress, and to identify factors related to mental health outcomes in health workers who had direct contact with patients with COVID-19 in hospitals in the southern region of Vietnam during the period of July–August 2020 when the second 'wave' of COVID-19 was occurring in Vietnam, mainly in the central region.

## 'Waves' of COVID-19 in Vietnam

The early phases of the COVID-19 pandemic in Vietnam have been unique compared to other countries. In a country of 97 million people [13], there have been four main periods of infection and containment, or 'waves', of COVID-19, including cases in the community and cases imported from abroad, with periods of relative 'normality' in between. The first wave was between January and July 2020 and resulted in 415 patients with no deaths recorded, the second wave was from July 2020 to January 2021 and resulted in 1,136 patients and 35 deaths but was mostly contained to the area surrounding Da Nang, the largest city in central Vietnam. The third wave occurred between January and April 2021 and was concentrated mostly in Hai Duong province in northern Vietnam. This wave resulted in 1,301 infections and no deaths. The current wave started in April 2021 and as of mid-August, has resulted in over 290,000 more infections and 6,440 deaths to date and it is still ongoing at the time of this publication [14].

We recruited participants, including physicians, nurses, and other health related staff from 14 hospitals that treated patients with COVID-19, in the southern region of Vietnam, including 6 hospitals in Ho Chi Minh City and 8 hospitals at the provincial and district levels. During the period when the study took place, there were 89 COVID-19 patients within these specific hospitals. The other patients in these settings were in quarantine as potential but not confirmed cases. We invited everyone from the 14 hospitals who ever had direct contact with COVID-19 patients to participate in the study. Direct contact was defined as face-to-face care of a COVID-19 patient, such as examination, providing injections, taking blood samples, oropharyngeal smear sampling, vital sign taking, and communication with the patients.

We designed a descriptive cross-sectional self-administered survey including sections on general information, symptoms experienced during patient care, a depression, anxiety and stress assessment, a post-traumatic stress assessment, and other related factors [4, 15–18]. Participants completed the questionnaire in Vietnamese on paper forms and then it was collected by a researcher who checked it for completeness. Those who were in isolation at the hospital due to treating patients with COVID-19 at that time still completed the questionnaire but it was not possible to obtain any missing data when collecting the forms. The research team followed the routine prevention measures within all hospital settings (e.g. wearing masks, physical distancing, health declarations).

In this paper we present the analysis of all components except the post-traumatic stress assessment. We used the 21 question Depression, Anxiety, and Stress Scales (DASS-21) [19], which has been validated in Vietnam [20]. Using the validated scoring methods, the average scores for depression, anxiety and stress were calculated by summing the scores and multiplying by 2. The cut-off scores for the symptoms of depression, anxiety and stress were 9, 7 and 14, respectively [19]. Participants in the study were considered to have symptoms of depression, anxiety and stress when their average scores were higher than the upper limit score.

We included variables to explore factors related to depression, anxiety and stress including basic demographic characteristics (e.g. age, sex, ethnicity, marital status, income), work related characteristics (e.g. education level, profession) and questions related to medical history (i.e. co-morbid health conditions), as well as COVID-19 related symptoms experienced during patient care during the time frame. The variables also included self-worry of transmitting COVID-19 to family and friends (normal/not worried/not so worried/worried/very worried), being shunned and stigmatized by friends, family, and community (yes, no), family being shunned and stigmatized by the community (yes, no), anxiety about media communication related to the COVID-19 pandemic (normal/not worried/not very worried/worried/very

worried), working hours during COVID-19 outbreak (≤8 hours, >8 hours), receiving psychological counseling and support during the care and treatment of COVID-19 patients (yes/no), and training for infection control taking care of COVID-19 patients (yes/no).

Data analysis was performed using STATA statistical software version 13.0. We used the frequency, percentage, median, interquartile ranges (IQRs) to describe the participants' characteristics, medical histories, and physical symptoms. To determine factors related to symptoms of depression, anxiety and stress, crude and multivariable logistic regression analysis was performed. The associations between risk factors and outcomes are presented as odds ratios (ORs) and 95% confidence intervals (CIs). Multivariable models included adjustment for confounders, including age and education. The significance level was set at $\alpha < 0.05$.

The study was approved by the Ethics Committee of the Ho Chi Minh City Institute of Public Health (Decision No. 610/QD-VYTCC, dated July 10, 2020). All participants provided written informed consent prior to participating in the survey. The survey was anonymous, i.e. no identifying information was collected. Participants were provided the right to withdrawal from the study at any time.

## Results

The survey was completed by 499 health workers who had direct contact with COVID-19 patients from July to September 2020, however, five participants were excluded from analysis due to missing data in the questionnaire (i.e. over 30% missing). The five excluded participants completed the survey when they were in isolation and therefore the researchers could not cross-check for completeness.

### Demographic, comorbid health conditions, and physical symptoms

The majority of participants were ≤ 40-years old (420/494, 85.0%), 51.6% were female, 92.9% identified as Kinh ethnicity (459/494), a little over half of the participants (55.3%) were married (273/494), and 56.1% had university and/or post-graduate degrees (277/494). Nurses accounted for the majority (57.9%) of those who participated. Almost half of the participants (224/494) worked more than 8 hours/day during the period. The median monthly personal income of study subjects was approximately 8 (5–12) million Vietnamese Dong or $348 ($217-$522). See Table 1.

Reported co-morbid health conditions were low. There were 21/494 (4.3%) participants with hypertension, 19/494 (3.9%) with hyperlipidemia, and 8/494 (1.6%) with cardiovascular conditions. The physical symptoms experienced by the participants during the study period included insomnia (89/494, 18%), fatigue (73/494, 14.8%), loss of appetite (61/494, 12.4%), headache (56/494, 11.3%), sore throat (21/494, 4.3%) and joint pain (22/494, 4.4%). The percentage of health workers who had COVID-19 testing was 24.7% for the period from March 2020 until the time that they completed the survey. No participants tested positive. See Table 2.

### Depression, anxiety, and stress among participants

Eighty-nine of 494 (18.0%) participants had symptoms of depression, 57 of 494 (11.5%) had symptoms of anxiety, and 38 of 494 (7.7%) participants experienced symptoms of stress. See Fig 1. Most health workers had mental health symptoms at mild and moderate levels. Only 2 to 4 cases of the total number of participants (n = 494) had severe or very severe symptoms, accounting for only 0.4–0.8% of the sample. See Fig 2.

**Table 1. Health care workers' demographic characteristics in Vietnam (n = 494).**

| Characteristics | Frequency (n) | Percentage (%) |
|---|---|---|
| Age | | |
| ≤ 30 | 210 | 42.5 |
| 31–40 | 210 | 42.5 |
| > 40 | 74 | 15.0 |
| Sex | | |
| Male | 239 | 48.4 |
| Female | 255 | 51.6 |
| Ethnicity | | |
| Kinh | 459 | 92.9 |
| Other (Tày, Chăm, Mường) | 35 | 7.1 |
| Marital status | | |
| Single | 211 | 42.7 |
| Married | 273 | 55.3 |
| Divorced/widowed | 10 | 2.0 |
| Education level | | |
| Intermediate/college | 217 | 43.9 |
| University/graduate | 277 | 56.1 |
| Profession | | |
| Doctor | 110 | 22.3 |
| Nurse | 286 | 57.9 |
| Technician (laboratory, radiologist) | 98 | 19.8 |
| Hours of work | | |
| ≤8 hours/day | 270 | 54.7 |
| >8 hours/day | 224 | 45.3 |
| Monthly income (million VND) Median (IQR) | 8 (5–12) | |

## Factors associated with depression, anxiety, and stress

The results of logistic regression analysis, after controlling for confounding factors including age group and education, showed several risk factors related to depression, anxiety, and stress, as well as some protective factors. First, health workers who worked for more than 8 hours per day during a shift during the COVID-19 pandemic period had higher odds for mental health issues than those who worked ≤ 8 hours per day (depression OR = 1.9, 95% CI (1.2–3.0), anxiety OR = 2.4, 95% CI (1.3–4.2) and stress OR = 3.5, 95% CI (1.6–7.4)). Participants who reported physical symptoms including sore throat, diarrhea, runny nose, cough, fatigue, muscle pain, headaches, insomnia and loss of appetite also had higher odds of psychological problems than those who did not report those symptoms. In addition, health workers with symptoms of joint pain had higher odds of depression and anxiety than those without these symptoms (depression OR = 5.3, 95%CI (2.0–13.9), anxiety OR = 4.4, 95% CI (1.6–11.6)), but not for stress (OR = 2.7, 95%CI (0.7–10.4)). Health workers who felt worried or very worried about transmitting COVID-19 to family and friends were more likely to experience mental health issues compared to those who felt normal/not worried/not so worried (depression OR = 4.2, 95%CI (1.9–9.0), anxiety OR = 2.6, 95% CI (1.2–5.9), and stress OR = 4.5, 95% CI (1.4–14.9)), and those who felt that they were shunned and stigmatized by friends, family and community members had higher odds of mental health problems than those who did not report stigma (depression OR = 3.1, 95%CI (1.9–5.0), anxiety OR = 3.0, 95% CI (1.7–5.3), and stress OR = 3.6, 95% CI (1.8–7.2)). Participants who stated that their families were shunned

**Table 2. Health care workers' medical history and physical symptoms (n = 494).**

| Clinical history and symptoms | Frequency (n) | Percentage (%) |
|---|---|---|
| Clinical history | | |
| Hypertension | 21 | 4.3 |
| Diabetes | 3 | 0.6 |
| Hyperlipidemia | 19 | 3.9 |
| Cardiovascular disease | 8 | 1.6 |
| Asthma | 5 | 1.0 |
| Mental illness | 1 | 0.2 |
| Physical symptoms | | |
| Fever $\geq 38,5\degree C$ | 1 | 0.2 |
| Sore throat | 21 | 4.3 |
| Runny nose | 18 | 3.6 |
| Cough | 19 | 3.9 |
| Sputum | 8 | 1.6 |
| Trouble breathing | 3 | 0.6 |
| Nausea, vomit | 4 | 0.8 |
| Diarrhea | 10 | 2.0 |
| Fatigue | 73 | 14.8 |
| Joint pain | 22 | 4.4 |
| Muscle pain | 25 | 5.1 |
| Headache | 56 | 11.3 |
| Insomnia | 89 | 18.0 |
| Loss of appetite | 61 | 12.4 |
| Itchy, skin rashes | 9 | 1.8 |
| COVID-19 Testing | | |
| Not tested | 372 | 75.3 |
| Tested | 122 | 24.7 |

and stigmatized had higher odds of mental health problems compared to those who did not report discrimination against their families (depression OR = 3.2, 95% CI (1.8–5.8), anxiety OR = 3.5, 95% CI (1.9–6.3), and stress OR = 4.3, 95% CI (2.1–8.7)). Finally, those who felt worried or very worried when watching the media about the COVID-19 pandemic were more

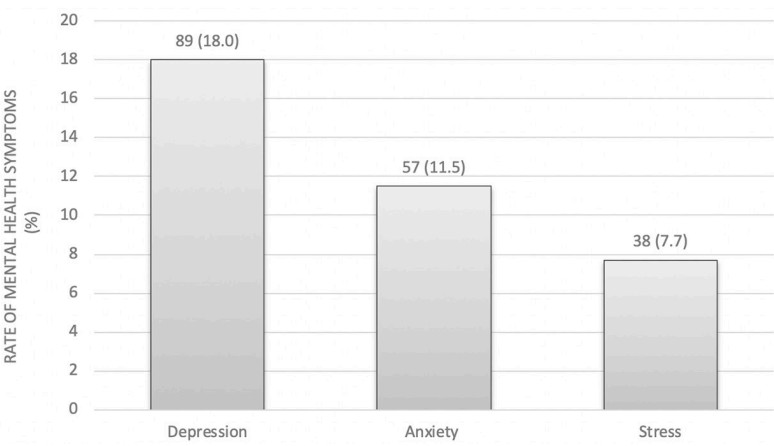

**Fig 1. The rates of depression, anxiety and stress among health workers.**

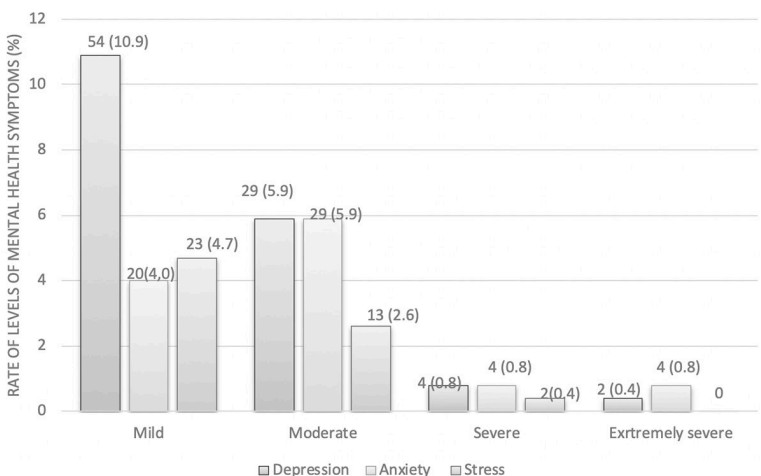

**Fig 2. Number of participants experiencing adverse psychological impact, stratified by severity.**

likely to be depressed (OR = 3.5, 95%CI (1.8–7.1)) compared to those who reported feeling normal/not worried/not so worried about media. However, worry related to media was not associated with anxiety and stress (anxiety OR = 1.8, 95% CI (0.9–3.7), stress OR = 1.5, 95% CI (0.7–3.4)). See Table 3.

For protective factors, health workers who received psychological counseling and felt supported during the care and treatment of COVID-19 patients were less likely to experience depression OR = 0.5, 95%CI (0.3–0.8), anxiety OR = 0.4, 95% CI (0.2–0.7), and stress OR = 0.4, 95% CI (0.2–0.7)). Finally, health workers who were trained on infection control prevention prior to care and treatment of patients with COVID-19 had decreased odds of depression (OR = 0.4, 95% CI (0.1–0.9), anxiety OR = 0.3, 95% CI (0.1–0.8), and stress OR = 0.2, 95% CI (0.1–0.5)) compared to the health workers who were not trained. See Table 3.

## Discussion

In this study, we determined the rates of depression, anxiety and stress among health workers who had direct care responsibilities for patients with COVID-19 during the second wave of COVID-19 in hospitals in southern Vietnam. The depression, anxiety and stress rates were 18.0%, 11.5% and 7.7% respectively. Overall, frontline health workers included in our study had lower rates of depression, anxiety and stress relative to other contexts with more severe outbreak patterns and most participants in our study experienced mild or moderate symptoms. Throughout the pandemic to date, mental health outcomes of health workers have varied greatly by context. In contexts with uncontrollable spread, early in the pandemic, there tended to be higher levels of mental health issues for health workers. For example, a study conducted in China from January to February 2020, showed that 50.4% of staff had symptoms of depression, 44.6% had symptoms of anxiety, 34% had symptoms of insomnia, and 71.5% had symptoms of distress [4]. At this time, the pandemic in China was extremely complicated, medical facilities were overloaded, and there was a major lack of human resources and protective equipment. According to reports as of 8/2/2020 in China, more than 34,878 people were already infected with COVID-19 and 724 people had died. In Hubei alone, there were 24,953 infections and 699 corona-related deaths [21]. Shortly thereafter, Rossi et al (2020) conducted online survey research in Italy from March 27–31, 2020 with medical staff, using a variety of

**Table 3. Models of factors associated with depression, anxiety, and stress among health workers caring for patients with COVID-19 (n = 494).**

| Risk factors | Depression | | | Anxiety | | | Stress | | |
|---|---|---|---|---|---|---|---|---|---|
| | Cases with symptoms/total cases (%) | Crude OR (95% CI) | Adjusted OR (95% CI) | Cases with symptoms/total cases (%) | Crude OR (95% CI) | Adjusted OR (95% CI) | Cases with symptoms/total cases (%) | Crude OR (95% CI) | Adjusted OR (95% CI) |
| Age | | | | | | | | | |
| ≤ 30 years old | 44/210 (21.0) | 1 (Reference) | | 23/210 (10.9) | 1 (Reference) | | 19/210 (9.1) | 1 (Reference) | |
| 31–40 years old | 38/210 (18.1) | 0.8 (0.5–1.4) | | 27/210 (12.9) | 1.2 (0.7–2.2) | | 14/210 (6.7) | 0.7 (0.4–1.5) | |
| > 40 years old | 7/74 (9.5) | 0.4 (0.2–0.9) | | 7/74 (9.5) | 0.9 (0.3–2.1) | | 5/74 (6.8) | 0.7 (0.3–2.0) | |
| Educational level | | | | | | | | | |
| Intermediate/college | 31/217 (14.3) | 1 (Reference) | | 28/217 (12.9) | 1 (Reference) | | 8/217 (3.7) | 1 (Reference) | |
| University/graduate | 58/277 (20.9) | 1.6 (1.0–2.6) | | 29/277 (10.5) | 0.8 (0.5–1.4) | | 30/277 (10.8) | 3.2 (1.4–7.1) | |
| Working time during COVID-19 pandemic | | | | | | | | | |
| ≤ 8 hours/day | 36/270 (13.3) | 1 (Reference) | 1 (Reference) | 21/270 (7.8) | 1 (Reference) | 1 (Reference) | 10/270 (3.7) | 1 (Reference) | 1 (Reference) |
| >8 hours/day | 53/224 (23.7) | 2.0 (1.3–3.2) | 1.9 (1.2–3.0) | 36/224 (16.1) | 2.3 (1.2–4.0) | 2.4 (1.3–4.2) | 28/224 (12.5) | 3.7 (1.8–7.8) | 3.5 (1.6–7.4) |
| Physical symptoms | | | | | | | | | |
| Sore throat | | | | | | | | | |
| No | 76/743 (16.1) | 1 (Reference) | 1 (Reference) | 46/473 (9.7) | 1 (Reference) | 1 (Reference) | 29/473 (6.1) | 1 (Reference) | 1 (Reference) |
| Yes | 13/21 (61.9) | 8.5 (3.4–21.2) | 9.5 (2.6–24.6) | 11/21 (52.4) | 10.2 (4.1–25.3) | 11.4 (4.5–29.0) | 9/21 (42.9) | 11.5 (4.5–29.5) | 11.4 (4.3–30.5) |
| Diarrhea | | | | | | | | | |
| No | 83/484 (17.2) | 1 (Reference) | 1 (Reference) | 52/484 (10.7) | 1 (Reference) | 1 (Reference) | 34/484 (7.0) | 1 (Reference) | 1 (Reference) |
| Yes | 6/10 (60.0) | 7.2 (2.0–26.2) | 8.4 (2.2–32.5) | 5/10 (50.0) | 8.3 (2.3–29.7) | 9.6 (2.6–35.1) | 4/10 (40.0) | 8.8 (2.4–32.8) | 8.5 (2.2–33.4) |
| Runny nose | | | | | | | | | |
| No | 81/476 (17.0) | 1 (Reference) | 1 (Reference) | 50/476 (10.5) | 1 (Reference) | 1 (Reference) | 31/476 (6.5) | 1 (Reference) | 1 (Reference) |
| Yes | 8/18 (44.4) | 3.9 (1.5–10.2) | 4.5 (1.7–12.2) | 7/18 (38.9) | 5.4 (2.0–14.6) | 6.0 (2.2–16.5) | 7/18 (38.9) | 9.1 (3.3–25.2) | 10.3 (3.5–30.5) |
| Cough | | | | | | | | | |
| No | 79/475 (16.6) | 1 (Reference) | 1 (Reference) | 49/475 (10.3) | 1 (Reference) | 1 (Reference) | 33/475 (6.9) | 1 (Reference) | 1 (Reference) |
| Yes | 10/19 (52.6) | 5.6 (2.1–14.1) | 5.6 (2.2–14.3) | 8/19 (42.1) | 6.3 (2.4–16.5) | 6.6 (2.5–17.5) | 5/19 (26.3) | 4.8 (1.6–14.1) | 4.5 (1.5–13.5) |
| Fatigue | | | | | | | | | |
| No | 52/421 (12.4) | 1 (Reference) | 1 (Reference) | 39/421 (9.3) | 1 (Reference) | 1 (Reference) | 19/421 (4.5) | 1 (Reference) | 1 (Reference) |
| Yes | 37/73 (50.7) | 7.3 (4.2–12.6) | 7.7 (4.4–13.6) | 18/73 (24.7) | 3.2 (1.7–6.0) | 3.6 (1.9–6.8) | 19/73 (26.0) | 7.4 (3.7–14.9) | 6.8 (1.4–13.9) |
| Muscle pain | | | | | | | | | |
| No | 77/469 (16.4) | 1 (Reference) | 1 (Reference) | 50/469 (10.7) | 1 (Reference) | 1 (Reference) | 32/469 (6.8) | 1 (Reference) | 1 (Reference) |
| Yes | 12/25 (48.0) | 4.7 (2.1–10.7) | 5.8 (2.5–13.7) | 7/25 (28.0) | 3.3 (1.3–8.2) | 3.3 (1.3–8.4) | 6/25 (24.0) | 4.3 (1.6–11.6) | 5.8 (2.0–16.5) |
| Headache | | | | | | | | | |
| No | 66/438 (15.1) | 1 (Reference) | 1 (Reference) | 46/438 (10.5) | 1 (Reference) | 1 (Reference) | 27/438 (6.2) | 1 (Reference) | 1 (Reference) |
| Yes | 23/56 (41.1) | 3.9 (2.2–7.1) | 3.7 (2.0–6.8) | 11/56 (19.6) | 2.1 (1.01–4.3) | 2.3 (0.1–4.7) | 11/56 (19.6) | 3.7 (1.7–8.0) | 3.3 (1.5–7.1) |
| Insomnia | | | | | | | | | |
| No | 53/405 (13.1) | 1 (Reference) | 1 (Reference) | 38/405 (9.4) | 1 (Reference) | 1 (Reference) | 20/405 (4.9) | 1 (Reference) | 1 (Reference) |
| Yes | 36/89 (40.5) | 4.5 (2.7–7.5) | 4.5 (2.6–7.6) | 19/89 (21.4) | 2.6 (1.4–4.8) | 2.8 (1.5–5.3) | 18/89 (20.2) | 4.9 (2.5–9.7) | 4.5 (2.2–9.1) |
| Loss of appetite | | | | | | | | | |
| No | 60/433 (13.9) | 1 (Reference) | 1 (Reference) | 39/433 (9.0) | 1 (Reference) | 1 (Reference) | 23/433 (5.3) | 1 (Reference) | 1 (Reference) |
| Yes | 29/61 (47.5) | 5.6 (3.2–9.9) | 5.4 (3.0–9.7) | 18/61 (29.5) | 4.2 (2.2–8.0) | 4.8 (2.4–9.3) | 15/61 (24.6) | 5.8 (2.8–11.9) | 5.2 (2.5–10.9) |
| Joint pain | | | | | | | | | |
| No | 80/472 (17.0) | 1 (Reference) | 1 (Reference) | 50/472 (10.6) | 1 (Reference) | 1 (Reference) | 35/472 (7.4) | 1 (Reference) | 1 (Reference) |
| Yes | 9/22 (40.9) | 3.4 (1.4–8.2) | 5.3 (2.0–13.9) | 7/22 (31.8) | 3.9 (1.5–10.1) | 4.4 (1.6–11.6) | 3/22 (13.6) | 2.0 (0.6–6.9) | 2.7 (0.7–10.4) |

*(Continued)*

**Table 3.** (Continued)

| Risk factors | Depression | | | Anxiety | | | Stress | | |
|---|---|---|---|---|---|---|---|---|---|
| | Cases with symptoms/total cases (%) | Crude OR (95% CI) | Adjusted OR (95% CI) | Cases with symptoms/total cases (%) | Crude OR (95% CI) | Adjusted OR (95% CI) | Cases with symptoms/total cases (%) | Crude OR (95% CI) | Adjusted OR (95% CI) |
| Fear of transmitting COVID-19 to family and friends | | | | | | | | | |
| Normal/not worried/not so worried | 8/124 (6.5) | 1 (Reference) | 1 (Reference) | 7/124 (5.7) | 1 (Reference) | 1 (Reference) | 3/124 (2.4) | 1 (Reference) | 1 (Reference) |
| Worried/very worried | 81/370 (21.9) | 4.1 (1.9–8.7) | 4.2 (1.9–9.0) | 50/370 (13.5) | 2.6 (1.2–5.9) | 2.6 (1.1–5.9) | 35/370 (9.5) | 4.2 (1.3–13.9) | 4.5 (1.4–14.9) |
| Being shunned and stigmatized by friends, family, and community | | | | | | | | | |
| No | 42/340 (12.4) | 1 (Reference) | 1 (Reference) | 26/243 (7.7) | 1 (Reference) | 1 (Reference) | 16/340 (4.7) | 1 (Reference) | 1 (Reference) |
| Yes | 47/154 (30.5) | 3.1 (1.9–5.0) | 3.1 (1.9–5.0) | 31/154 (20.1) | 3.0 (1.7–5.3) | 3.0 (1.7–5.3) | 22/154 (14.3) | 3.4 (1.7–6.6) | 3.6 (1.8–7.2) |
| Family being shunned and stigmatized by the community | | | | | | | | | |
| No | 55/396 (14.0) | 1 (Reference) | 1 (Reference) | 33/396 (8.3) | 1 (Reference) | 1 (Reference) | 21/396 (5.3) | 1 (Reference) | 1 (Reference) |
| Yes | 34/98 (34.7) | 3.3 (2.0–5.5) | 3.2 (1.8–5.8) | 24/98 (24.5) | 3.6 (2.0–6.4) | 3.5 (1.9–6.3) | 17/98 (17.4) | 3.7 (1.9–7.4) | 4.3 (2.1–8.7) |
| Anxiety when seeing media communication on COVID-19 pandemic | | | | | | | | | |
| Normal/not worried/not very worried | 10/134 (7.5) | 1 (Reference) | 1 (Reference) | 10/134 (7.5) | 1 (Reference) | 1 (Reference) | 8/134 (6.0) | 1 (Reference) | 1 (Reference) |
| Worried/very worried | 79/360 (21.9) | 3.5 (1.7–6.9) | 3.5 (1.8–7.1) | 47/360 (13.1) | 1.9 (0.9–3.8) | 1.8 (0.9–3.7) | 30/360 (8.3) | 1.4 (0.6–3.2) | 1.5 (0.7–3.4) |
| Health worker received psychological counseling and supported during the care and treatment of COVID-19 patients | | | | | | | | | |
| No | 42/156 (26.9) | 1 (Reference) | 1 (Reference) | 27/156 (17.3) | 1 (Reference) | 1 (Reference) | 22/156 (14.1) | 1 (Reference) | 1 (Reference) |
| Yes | 47/338 (13.9) | 0.4 (0.3–0.7) | 0.5 (0.3–0.8) | 30/338 (8.9) | 0.5 (0.3–0.8) | 0.4 (0.2–0.7) | 16/338 (4.7) | 0.3 (0.1–0.6) | 0.4 (0.2–0.7) |
| Training for infection control taking care of COVID-19 patients | | | | | | | | | |
| No | 7/20 (35.0) | 1 (Reference) | 1 (Reference) | 6/20 (30.0) | 1 (Reference) | 1 (Reference) | 5/20 (25.0) | 1 (Reference) | 1 (Reference) |
| Yes | 82/474 (17.3) | 0.4 (0.2–1.0) | 0.4 (0.1–0.9) | 51/474 (10.8) | 0.3 (0.1–0.8) | 0.3 (0.1–0.8) | 33/474 (7.0) | 0.2 (0.1–0.7) | 0.2 (0.1–0.5) |

questionnaires, and found rates of severe depression, anxiety, and severe stress were 24.73%, 19.80%, and 21.9%, respectively [22]. Many contexts found high rates. For example, according to Nayak et al (2021) the rates of depression, anxiety and stress in HCWs in Trinidad and Tobago were 42.28%, 56.2% and 17.97%, respectively [23]. However in Singapore, researchers conducted a study between February and March 2020, also using DASS 21, and found lower rates of depression, anxiety and stress: 8.9%, 14.5% and 6.6%, respectively [15].

Results from a study conducted with 173 health workers in two national hospitals in Hanoi, Vietnam during the first wave (March to April 2020) found that 20.2%, 33.5%, and 12% of health workers experienced depression, anxiety, and stress, respectively, and this was significantly higher for those working in COVID-19 designated hospitals [24]. These rates were higher than the rates that we found in our study, although our study was conducted several months after the initial wave. There were also two other studies conducted in Vietnam at a similar time as our study, with higher rates. Nhan et al (2021) conducted a study in Da Nang from July to September 2020 and found stress rates among frontline healthcare workers at 44.6%, with 18.9% of those experiencing severe or extremely severe stress [25]. Tuan et al (2021) also conducted a study in Da Nang and Quang Nam province in August 2020 and found the prevalence of anxiety and depression was 26.84% and 34.70%, respectively [26]. Da Nang and Quang Nam were the two hotspots with the highest number of COVID patients in

the second wave occurring in Vietnam, while the southern region where our study took place had fewer cases and most of the positive cases were from abroad and had already been isolated.

We also found several risk factors for depression, anxiety and stress. Our results showed that health workers who worked over 8 hours/day during this period had increased odds of depression, anxiety, and stress than those who worked ≤ 8 hours/day (depression OR = 1.9, 95% CI (1.2–3.0), anxiety OR = 2.4, 95% CI (1.3–4.2) and stress OR = 3.5, 95% CI (1.6–7.4)). Similarly, Huang et al (2020) found that health workers in China who treated COVID-19 patients for more than 3 hours a day had higher rates of anxiety and depression than those who worked less time [16] and Ahn et al (2020) in Korea also found that working time is a factor significantly related to depression [27]. Tuan et al (2021) also showed that working more hours per week increased the odds of stress by 12‰ (OR = 1.012; 95% CI: 1.004–1.019) [26].

For health workers who experienced several physical symptoms (including sore throat, diarrhea, runny nose, cough, fatigue, muscle pain, headaches, insomnia and loss of appetite), there was an association with depression, anxiety, and stress. Only 24.7% of the participants in our study ever had COVID-19 testing since the start of the pandemic, so many health workers experiencing symptoms did not have immediate confirmation regarding whether or not the symptoms were related to COVID-19 or not. Similarly, Chew et al. (2020) found that the presence of physical symptoms (such as sore throat, vomiting, nausea, insomnia, loss of appetite, etc.) was associated with a higher mean score on the health-care workers' scales of anxiety, stress, and depression [18]. However, because our study was a cross-sectional study, we do not know if the physical symptoms experienced are a cause or consequence of the mental health issues.

There were also community level factors that impacted depression, anxiety and stress of health workers in our setting. Health workers who felt worried or very worried about transmitting COVID-19 to family and friends had higher odds of experiencing depression, anxiety and stress. According to a study by Maunder et al. (2003) in Canada during the SARS epidemic, health workers who cared for patients with SARS had a constant fear of being infected and transmitting SARS to their family, friends and the wider community, which was accompanied by a wider fear of stigma [9]. Similar results were found during COVID-19 where Nguyen et al (2021) found that being exposed to COVID-19 at work and taking the infection home to their family is a contributing factor to stress of both medical and non-medical HCWs [28]. In our research, we also found that the health workers who were shunned by friends, family, community, and whose families were stigmatized by the community were more likely to experience depression, anxiety and stress. Studies conducted by Sritharan et al (2020) and Greene et al (2021) have both shown that stigma was one of the risk factors for decreased mental health outcomes of frontline workers [29, 30].

Additionally, health workers in our study who felt worried or very worried when watching the media about the COVID-19 pandemic were more likely to be depressed than those who reported they felt normal/not worried/not so worried. This result was similar with the study of Gao et al. (2020) in China, which showed that regular media exposure during the COVID-19 outbreak in Wuhan was strongly associated with the prevalence of depression and anxiety (OR = 1.91, 95% CI: 1.52–2.41) [31].

The protective factors we found in this study are worth mentioning. First, those who received psychological counseling and support during the care and treatment of COVID-19 patients had reduced odds of depression, anxiety and stress. Secondly, health workers who were trained on infection control prevention prior to care and treatment of patients with COVID-19 had decreased risks of depression, anxiety and stress compared to those who were not trained. Both Maunder et al. (2006) and Lancee et al. (2008) found that adequate training for medical staff was a potential protective factor in preventing post-traumatic stress disorders

during the SARS pandemic [9, 32]. Similarly, Tuan et al (2021) also showed that an increased score of HCWs' knowledge of COVID-19 reduced the odds of experiencing stress (OR = 0.853; 95% CI: 0.739–0.986) [26].

The study has several limitations. First, this was a cross-sectional study, so we are not able to identify the direction of causality in associations between risk factors and mental health outcomes. As Chew et al (2020) report, these effects are likely to be bi-directional [18]. Second, in this analysis, we only explored health workers working in hospitals in a specific region and during a specific time point during the COVID-19 pandemic, so the results may not represent the mental health outcomes of health workers throughout Vietnam or during different phases of the pandemic. We also did not include participants who were in management positions or related health roles that did not include direct patient contact within these institutions. Their experiences and perspectives may be different. Finally, we did not collect qualitative data to explore the meaning behind concepts, for example, feeling shunned or experiencing stigma.

In conclusion, the rates of depression, anxiety and stress of health workers taking care of patients with COVID-19 at hospitals in southern Vietnam were lower than in other countries experiencing more uncontrolled COVID-19 transmission. Further exploration of the association between physical symptoms experienced by health workers and mental health may guide interventions to improve health outcomes. More routine COVID-19 testing among health workers could reduce anxieties about physical symptoms and alleviate the fear of transmitting COVID-19 to family and friends. Two important protective factors that could help to reduce the risk for depression, anxiety and stress among health workers include psychological counseling and support, and training in infection prevention. Medical institutions need to ensure that health workers have access to basic trainings, prior to initiation of work, and routine mental health support during the pandemic and into the future. The results of this study provide a basis for hospital administrative staff and local and national governments to develop appropriate plans, policies and interventions to protect the mental health of health workers working in pandemic contexts.

## Supporting information

**S1 Questionnaire. English-language questionnaire.**
(DOCX)

**S2 Questionnaire. Vietnamese-language questionnaire.**
(DOCX)

**S1 Data.**
(DTA)

## Acknowledgments

We would like to express our gratitude to the Centers for Disease Control and Prevention of the United States (US-CDC), South Asian Field Epidemiology and Technology Network, (SAFETYNET), and the Field Epidemiology Training Program (FETP) in Vietnam for their generous funding for this study. We also sincerely thank the Boards of Directors and the health workers of the 14 hospitals in southern Vietnam for supporting and collaborating with us on this research.

## Author Contributions

**Conceptualization:** Anh Le Thi Ngoc, Chinh Dang Van, Phong Nguyen Thanh.

**Formal analysis:** Anh Le Thi Ngoc, Sonia Lewycka.

**Funding acquisition:** Anh Le Thi Ngoc, Chinh Dang Van, Phong Nguyen Thanh.

**Methodology:** Anh Le Thi Ngoc, Phong Nguyen Thanh.

**Project administration:** Anh Le Thi Ngoc.

**Supervision:** Chinh Dang Van.

**Writing – original draft:** Anh Le Thi Ngoc, Jennifer Ilo Van Nuil.

**Writing – review & editing:** Anh Le Thi Ngoc, Chinh Dang Van, Phong Nguyen Thanh, Sonia Lewycka, Jennifer Ilo Van Nuil.

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
