## [Decision Letter · Decision Letter 0]

8 Feb 2022

PGPH-D-21-00531

Depression, anxiety, and stress among frontline health workers during the second wave of COVID-19 in southern Vietnam: a cross-sectional survey

Dear Dr. Van Nuil,

Thank you for submitting your manuscript to PLOS Global Public Health. After careful consideration, we feel that it has merit but does not fully meet PLOS Global Public Health’s publication criteria as it currently stands. Therefore, we invite you to submit a revised version of the manuscript that addresses the points raised during the review process.

We note that one or more reviewers has recommended that you cite specific previously published works. As always, we recommend that you please review and evaluate the requested works to determine whether they are relevant and should be cited. It is not a requirement to cite these works. We appreciate your attention to this request.

We look forward to receiving your revised manuscript.

Kind regards,

Marianne Clemence, Staff Editor, PLOS ONE, on behalf of,

Behdin Nowrouzi-Kia

Academic Editor

Journal Requirements:

1. Please provide additional details regarding participant consent. In the ethics statement, please ensure that you have specified whether consent was informed.

3. Please provide separate figure files in .tif or .eps format only.  Please ensure that all files are under our size limit of 20MB.  

For more information about how to convert your figure files please see our guidelines: Once you've converted your files to .tif or .eps, please also make sure that your figures meet our format requirements

4. In the online submission form, you indicated that "The dataset used and analysed during the current study are available from the corresponding author on reasonable request.". All PLOS journals now require all data underlying the findings described in their manuscript to be freely available to other researchers, either 1. In a public repository, 2. Within the manuscript itself, or 3. Uploaded as supplementary information.

5. Please amend your detailed Financial Disclosure statement. This is published with the article, therefore should be completed in full sentences and contain the exact wording you wish to be published.

ii). State the initials, alongside each funding source, of each author to receive each grant.

iii). State what role the funders took in the study. If the funders had no role in your study, please state: “The funders had no role in study design, data collection and analysis, decision to publish, or preparation of the manuscript.”

Additional Editor Comments (if provided):

Thank you for submitting your manuscript. We have now received the reviewers revisions and invite you to prepare a revised manuscript based on their comments.

Reviewers' comments:

Reviewer's Responses to Questions

**Comments to the Author**

1. Does this manuscript meet PLOS Global Public Health’s publication criteria? Is the manuscript technically sound, and do the data support the conclusions? The manuscript must describe methodologically and ethically rigorous research with conclusions that are appropriately drawn based on the data presented.

Reviewer #1: Yes

Reviewer #2: Yes

Reviewer #3: Yes

Reviewer #4: Yes

Reviewer #5: Yes

2. Has the statistical analysis been performed appropriately and rigorously?

Reviewer #1: Yes

Reviewer #2: Yes

Reviewer #3: Yes

Reviewer #4: Yes

Reviewer #5: Yes

3. Have the authors made all data underlying the findings in their manuscript fully available (please refer to the Data Availability Statement at the start of the manuscript PDF file)?

Reviewer #1: Yes

Reviewer #2: Yes

Reviewer #3: Yes

Reviewer #4: Yes

Reviewer #5: Yes

4. Is the manuscript presented in an intelligible fashion and written in standard English?

Reviewer #1: Yes

Reviewer #2: Yes

Reviewer #3: Yes

Reviewer #4: Yes

Reviewer #5: Yes

5. Review Comments to the Author

Reviewer #1: A very interesting and methodologically valid work. It is necessary to insert some bibliographical references that improve the paper and that better justify the methodological work done that I strongly suggest to insert:

Chirico F, Ferrari G. Role of the workplace in implementing mental health interventions for high-risk groups among the working age population after the COVID-19 pandemic. J Health Soc Sci. 2021;6(2):145-150. Doi: 10.19204/2021/rlft1.

Magnavita N, Chirico F, Garbarino S, Bragazzi NL, Santacroce E, Zaffina S. SARS/MERS/SARS-CoV-2 Outbreaks and Burnout Syndrome among Healthcare Workers. An umbrella Systematic Review. Int J Environ Res Public Health. 2021;18(8):4361. Doi: 10.3390/ijerph18084361.

Chirico F, Nucera G. Tribute to healthcare operators threatened by COVID-19 pandemic. J Health Soc Sci. 2020;5(2):165-168. 10.19204/2020/trbt1

Chirico F, Magnavita N. Burnout Syndrome and Meta-Analyses: Need for Evidence-Based Research in Occupational Health. Comments on Prevalence of Burnout in Medical and Surgical Residents: A Meta-Analysis. Int. J. Environ. Res. Public. Health. 2019, 16, doi:10.3390/ijerph16091479. Int J Environ Res Public Health. 2020;17(3):741. Published 2020 Jan 23. doi:10.3390/ijerph17030741

Babore, A., Lombardi, L., Viceconti, M. L., Pignataro, S., Marino,V., Crudele, M., . . . Trumello, C. (2020). Psychological effects of the COVID-2019 pandemic: perceived stress and coping strategies among healthcare professionals. Psychiatry Research. Retrieved from: https://www.ncbi.nlm.nih.gov/pmc/articles/PMC7397939/

Rana, W., Mukhtar S., & Mukhtar S. (2020). Mental health of medical workers in Pakistan during the pandemic COVID- 19 outbreak. Asian Journal of Psychiatry, 51, 102080. doi: 10.1016/j.ajp.2020.102080. 102080.

Reviewer #2: A very interesting article, robust in methodology and statistical analysis. Unfortunately, it is a bit lackluster in the introduction. For the publication it is necessary to increase the analysis of the literature with recent contributions and, dunnque, to expand the references. I strongly suggest the inclusion of the following articles:

Chirico F, Ferrari G, Nucera G, Szarpak L, Crescenzo P, Ilesanmi O. Prevalence of anxiety, depression, burnout syndrome, and mental health disorders among healthcare workers during the COVID-19 pandemic: A rapid umbrella review of systematic reviews. J Health Soc Sci. 2021;6(2):209-220. Doi: 10.19204/2021/prvl7.

Chirico F, Ferrari G. Role of the workplace in implementing mental health interventions for high-risk groups among the working age population after the COVID-19 pandemic. J Health Soc Sci. 2021;6(2):145-150. Doi: 10.19204/2021/rlft1.

Magnavita N, Chirico F, Garbarino S, Bragazzi NL, Santacroce E, Zaffina S. SARS/MERS/SARS-CoV-2 Outbreaks and Burnout Syndrome among Healthcare Workers. An umbrella Systematic Review. Int J Environ Res Public Health. 2021;18(8):4361. Doi: 10.3390/ijerph18084361.

Chirico F, Crescenzo P, Sacco A, Riccò M, Ripa S, Nucera G, Magnavita N. Prevalence of burnout syndrome among Italian volunteers of the Red Cross: a cross-sectional study. Ind Health. 2021;59(2):117-127. doi: 10.2486/indhealth.2020-0246.

Chirico F, Nucera G. Tribute to healthcare operators threatened by COVID-19 pandemic. J Health Soc Sci. 2020;5(2):165-168. 10.19204/2020/trbt1

Chirico F, Magnavita N. Burnout Syndrome and Meta-Analyses: Need for Evidence-Based Research in Occupational Health. Comments on Prevalence of Burnout in Medical and Surgical Residents: A Meta-Analysis. Int. J. Environ. Res. Public. Health. 2019, 16, doi:10.3390/ijerph16091479. Int J Environ Res Public Health. 2020;17(3):741. Published 2020 Jan 23. doi:10.3390/ijerph17030741

Crescenzo, P., Marciano, R., Maiorino, A, Denicolo, D., D’Ambrosi, D., Ferrara, I, Calabrese, S., & Diodato, F. (2021). First COVID-19 wave in Italy: coping strategies for the prevention and prediction of burnout syndrome (BOS) in voluntary psychologists employed in telesupport. Psychology Hub, 38(1), 31-38 doi: 10.13133/2724-2943/17435

Reviewer #3: In this study of stress, anxiety and depression in frontline HCWs of various Vietnamese hospitals during the second wave of COVID-19, the authors cited several references to support their findings. However, they failed to consider a longitudinal study that was conducted in the staff of a single COVID-19 hub-hospital, who were continuously exposed throughout the pandemic. This study allows us to observe the differences between the stress factors present in the first wave of the pandemic [Magnavita N, Soave PM, Ricciardi W, Antonelli M. Occupational stress and mental health of anaesthetists during the COVID-19 pandemic. Int J Environ Res Public Health 2020, 17, 8245; doi:10.3390/ijerph17218245], In the second phase [Magnavita N, Soave PM, Antonelli M. Prolonged Stress Causes Depression in Frontline Workers Facing the COVID-19 Pandemic-A Repeated Cross-Sectional Study in a COVID-19 Hub-Hospital in Central Italy. Int J Environ Res Public Health. 2021 Jul 8;18(14):7316. doi: 10.3390/ijerph18147316.] and in the third phase [Magnavita, N.; Soave, P.M.; Antonelli, M. One-Year Prospective Study of Occupational Health in the Intensivists of a COVID-19 Hub Hospital. Preprints 2021, 2021080423 (doi: 10.20944/preprints202108.0423.v1)]. The high intensity of the occupational effort in these workers explains the higher prevalence of disorders compared to those of the Vietnamese experience.

Another experience that the authors could consider is that which involved all workers of a local health unit in Italy during the first wave of the pandemic, in April 2020 [Magnavita N, Tripepi G, Di Prinzio RR. Symptoms in Health Care Workers during the COVID-19 Epidemic. A Cross-Sectional Survey. Int J Environ Res Public Health. 2020;17(14):5218. doi: 10.3390/ijerph17145218.]. In this experience the overall prevalence of anxiety and depression was not greater than as recorded in previous years in the same population, but workers who had unprotected exposure and even more those who had a positive nasopharyngeal test for SARS-CoV-2 showed a significantly increased risk of anxiety and depression.

Reviewer #4: 1. Line 68: Perhaps clearer connection can be build between this paragraph and paragraph before.

2. Please include ethics clearance in the methodology section. Please move Line 132 and Line 105.

3. Please include token and right for withdrawal in the methodology section.

4. Could you please further clarify how did researcher obtained clearance to collect the data inside the hospital where it's with loads of potential COVID-19 patients? Any precaution steps being taken?

5. What was the language used in the survey?

6. Line 113- why and how are these variables being included? Perhaps good to cite some references to support your selection of variables.

7. SPSS statistical software version 16.0- is bit too old since it's now with Version 23 and above. Please double check.

8. Line 90: We recruited participants from 14 hos 90 hospitals that treated patients with...Are they nurse or? Perhaps good to have some basic introduction of the health workers here? Just a simple brief line about who they maybe.

9. The results and data analysis: No major issues. Well done.

10. Please highlight and include the contribution of this study in the conclusion area.

Reviewer #5: This is an interesting study that looks at exploring anxiety, depressive & stress among HCW during 2nd wave of COVID pandemic. In general, the topic is of great interest, the methods are robust and the manuscript is well written.

6. PLOS authors have the option to publish the peer review history of their article (what does this mean?). If published, this will include your full peer review and any attached files.

**Do you want your identity to be public for this peer review?** For information about this choice, including consent withdrawal, please see our Privacy Policy.

Reviewer #1: No

Reviewer #2: No

Reviewer #3: **Yes: **Nicola Magnavita

Reviewer #4: No

Reviewer #5: **Yes: **Walaa Sabry

---

## [Decision Letter · Decision Letter 1]

30 Jun 2022

Depression, anxiety, and stress among frontline health workers during the second wave of COVID-19 in southern Vietnam: a cross-sectional survey

PGPH-D-21-00531R1

Dear Dr. Van Nuil,

We are pleased to inform you that your manuscript 'Depression, anxiety, and stress among frontline health workers during the second wave of COVID-19 in southern Vietnam: a cross-sectional survey' has been provisionally accepted for publication in PLOS Global Public Health.

Best regards,

Behdin Nowrouzi-Kia

Academic Editor

Reviewer Comments (if any, and for reference):

Reviewer's Responses to Questions

**Comments to the Author**

1. If the authors have adequately addressed your comments raised in a previous round of review and you feel that this manuscript is now acceptable for publication, you may indicate that here to bypass the “Comments to the Author” section, enter your conflict of interest statement in the “Confidential to Editor” section, and submit your "Accept" recommendation.

Reviewer #3: All comments have been addressed

Reviewer #4: All comments have been addressed

Reviewer #5: (No Response)

2. Does this manuscript meet PLOS Global Public Health’s publication criteria? Is the manuscript technically sound, and do the data support the conclusions? The manuscript must describe methodologically and ethically rigorous research with conclusions that are appropriately drawn based on the data presented.

Reviewer #3: Yes

Reviewer #4: Yes

Reviewer #5: Yes

3. Has the statistical analysis been performed appropriately and rigorously?

Reviewer #3: Yes

Reviewer #4: Yes

Reviewer #5: Yes

4. Have the authors made all data underlying the findings in their manuscript fully available (please refer to the Data Availability Statement at the start of the manuscript PDF file)?

Reviewer #3: Yes

Reviewer #4: Yes

Reviewer #5: Yes

5. Is the manuscript presented in an intelligible fashion and written in standard English?

Reviewer #3: Yes

Reviewer #4: Yes

Reviewer #5: Yes

6. Review Comments to the Author

Reviewer #3: The authors addressed the comments and improved the manuscript

Reviewer #4: The authors have provided sufficient evidence and updated the manuscript.

Reviewer #5: Am interesting well written article with sound methodology.

7. PLOS authors have the option to publish the peer review history of their article (what does this mean?). If published, this will include your full peer review and any attached files.

**Do you want your identity to be public for this peer review?** For information about this choice, including consent withdrawal, please see our Privacy Policy.

Reviewer #3: **Yes: **Nicola Magnavita

Reviewer #4: No

Reviewer #5: **Yes: **Walaa Sabry
